

# Towards a pragmatic use of statistics in ecology

Leonardo Braga Castilho[1] and Paulo Inácio Prado[2]

[1] Universidade de Brasília, Brasília, Brazil
[2] Departamento de Ecologia/Instituto de Biociências, Universidade de São Paulo, São Paulo, São Paulo, Brazil

## ABSTRACT

Although null hypothesis testing (NHT) is the primary method for analyzing data in many natural sciences, it has been increasingly criticized. Recently, approaches based on information theory (IT) have become popular and were held by many to be superior because it enables researchers to properly assess the strength of the evidence that data provide for competing hypotheses. Many studies have compared IT and NHT in the context of model selection and stepwise regression, but a systematic comparison of the most basic uses of statistics by ecologists is still lacking. We used computer simulations to compare how both approaches perform in four basic test designs (t-test, ANOVA, correlation tests, and multiple linear regression). Performance was measured by the proportion of simulated samples for which each method provided the correct conclusion (power), the proportion of detected effects with a wrong sign (S-error), and the mean ratio of the estimated effect to the true effect (M-error). We also checked if the $p$-value from significance tests correlated to a measure of strength of evidence, the Akaike weight. In general both methods performed equally well. The concordance is explained by the monotonic relationship between $p$-values and evidence weights in simple designs, which agree with analytic results. Our results show that researchers can agree on the conclusions drawn from a data set even when they are using different statistical approaches. By focusing on the practical consequences of inferences, such a pragmatic view of statistics can promote insightful dialogue among researchers on how to find a common ground from different pieces of evidence. A less dogmatic view of statistical inference can also help to broaden the debate about the role of statistics in science to the entire path that leads from a research hypothesis to a statistical hypothesis.

Corresponding author
Leonardo Braga Castilho,
leonardobcastilho@gmail.com

## INTRODUCTION

Null hypothesis testing (NHT) has been the primary statistical method for drawing conclusions from data in natural sciences since the mid-1920s (*Huberty, 1993*). The purpose of NHT was originally to protect researchers from taking noise as a true effect (*Mayo & Spanos, 2011*; *Gelman & Carlin, 2014*). The probability of making such a mistake is gauged by the $p$-value calculated from a model of absence of effects (the null hypothesis). Accordingly, a model-driven definition of $p$-values was recently provided by

the American Statistical Society: *"the probability under a specified statistical model that a statistical summary of the data (…) would be equal to or more extreme than its observed value"* (*Wasserstein & Lazar, 2016*). However, *p*-values are frequently misinterpreted as evidence about the models themselves (*Cohen, 1994*; *Royall, 2000*). The most common misunderstandings are taking *p*-values as the probability that the null hypothesis is true, as how improbable the alternative hypothesis is, or as a measure of the effect strength (*Cohen, 1994*; *Greenland et al., 2016*; *Wasserstein & Lazar, 2016*). Thus, in a broad sense, NHT is an error-control procedure that has been widely misused to express the support data provides for a given hypothesis or model. Although this criticism is not new (see *Cohen, 1994*), it has been recently used to challenge not only NHT, but the body of scientific knowledge that has been acquired using NHT (*Ioannidis, 2005*; *Nuzzo, 2014*).

More recently the information theoretic (IT) approach has been vigorously championed as a response to these problems. Recent reviews have presented IT as the proper approach to assess the support data gives to competing models, and popularized the criticisms to the NHT among biologists (*Burnham & Anderson, 2002*; *Johnson & Omland, 2004*). Within the IT framework, one elaborates multiple hypotheses about the problem at hand and proposes a statistical model to express each one. A measure of the relative information loss resulting of each model is then used to identify which models are the best approximations of the data. The central concept is likelihood, which is any function proportional to the probability that a model assigns to the data. The likelihood function expresses the degree to which the data supports each model, and inference in IT is used to find the best supported model (*Edwards, 1972*; *Burnham & Anderson, 2002*). Although this approach is more used in the framework of model selection, *Burnham & Anderson (2002)* advocate that it has a much broader range of uses, encompassing all types of NHT analyses done today. The authors state that NHT is a poor method of analyzing data and strongly emphasize the use of the IT approach for every type of analysis in ecology.

These ideas elicited an intense discussion on the advantages of substituting NHT by the IT approach in recent years. Many authors have compared both methods using real and simulated data (*Whittingham et al., 2006*; *Glatting et al., 2007*; *Murtaugh, 2009*; *Freckleton, 2011*; *Lukacs, Burnham & Anderson, 2010*), and raised philosophical and theoretical issues (*Johnson & Omland, 2004*; *Steidl, 2006*; *Garamszegi et al., 2009*); some have also identified theoretical or usage problems arising from the IT approach (*Anderson & Burnham, 2002*; *Guthery et al., 2005*; *Lukacs et al., 2007*; *Mundry, 2011*; *Galipaud et al., 2014*), while others highlighted the importance of maintaining both approaches in practice (*Richards, 2005*; *Stephens et al., 2005*; *Stephens et al., 2007*).

Most of these works, however, have focused on comparing the IT approach to stepwise regression, a traditional technique for model selection in the NHT approach. Nevertheless, there is a long history of studies on the shortcomings of stepwise regression (*Quinn & Keough, 2002*), and many traditional problems in NHT, particularly in stepwise regression, also appear when using the IT approach (*Hegyi & Garamszegi, 2011*; *Mundry, 2011*). Proponents of the new framework based on the IT approach, however, do not advocate the use of the method only as a substitute for traditional stepwise regression, or for more

complex modeling situations. Instead, they view NHT as a whole as a poor method with much less inferential power than the IT approach (*Anderson, 2008*). To compare the IT approach to a technique already shown to be poor, as has been done for stepwise regression, is not a productive way to evaluate the advantages of the proposed new framework, if there are any. Furthermore, model selection is the home territory of IT approaches, while a less appreciated topic is the use of the *p*-value to reject null hypotheses in standard NHT designs such as t-tests, ANOVA or linear regression. In these paradigmatic cases, *p*-values are functions of the IT measures of evidence (*Edwards, 1972*; *Murtaugh, 2014*), suggesting that both approaches would lead to the same conclusions. However, as both approaches rely on asymptotic theories (*Aho, Derryberry & Peterson, 2017*), we need to check their congruence with the sample sizes usual for each knowledge area. The analytic correspondence of the IT and NHT approaches reveal important differences in asymptotic convergence in some simple cases (Fig. S1). For more complicated designs and realistic situations and also to estimate type-S and type-M errors (*Gelman & Carlin, 2014*, see below), computer simulations offer a straightforward way to make such comparisons.

Meanwhile, despite all the criticisms, NHT is still widely used and taught in many research disciplines, including ecology (*Stanton-Geddes, De Freitas & De Sales Dambros, 2014*; *Touchon & McCoy, 2016*; *Wasserstein & Lazar, 2016*). By progressing while using a technique that is the subject of many criticisms, researchers might be demonstrating that they can achieve their goals without worrying too much about philosophical controversies (*Mayo & Cox, 2006*). Practicing scientists might therefore feel they can agree on the conclusions drawn from a data set even if they use different statistical approaches (*e.g. Silberzahn et al., 2018*). One obvious reason for such a pragmatic agreement is the equivalence of the conclusions using the NHT and IT approaches. The aim of this study is to test such agreement using computer simulations to run pragmatic comparisons of the NHT and the IT methodologies in standard, realistic designs for ecological studies. Pragmatic criteria assign equivalence to any outcome of equal practical consequence, despite differences in the causes (*Hookway, 2016*). As with any phenomenological approach, pragmatic conclusions are context-dependent and so the context must be clearly stated. Therefore, our main question is whether statistical approaches that differ in theory can lead to the same conclusion under the realistic conditions often seen in the field or in the laboratory and in cases in which both approaches are possible. Specifically, we asked if there is any difference in the conclusions drawn from data traditionally analyzed with t-tests, ANOVA, correlation tests, or linear regression when analyzed with the IT approach. We also assessed whether the use of *p*-values to express the strength of evidence of the conclusions led to incorrect evaluations of the support provided by the data.

Hereafter we will call a rightful or correct conclusion a result from a statistical analysis that accords with the true model. The probability of detecting such effects (power) is the usual means of gauging how frequently significance tests produce accurate conclusions. Nevertheless, a significant effect can still lead to a wrong conclusion because the estimated effect can have the opposite sign or an inflated magnitude in the sample. *Gelman & Carlin (2014)* defined these errors as *type-S* and *type-M* and showed that their rates

increase as the test power decreases. These two types of errors have recently gained more attention in ecology (*Lemoine et al., 2016*; *Cleasby et al., 2021*). We thus combined test power, type-S and type-M errors to evaluate the performance of the IT and NHT approaches in providing accurate conclusions regarding statistical effects.

## METHODS

We compared NHT and IT approaches for four standard analysis designs in ecology: (i) unpaired t-test design; (ii) single-factor ANOVA design; (iii) correlation design; and (iv) multiple linear regression design. For each of these designs, we sampled values from the distributions assumed by each design (univariate Gaussian for t-tests and ANOVA, and bi-variate Gaussian for correlation tests and multiple linear regressions, details below). We then performed NHT and IT procedures with the simulated samples and compared the results of each with regard to the probability of achieving a correct conclusion, and the magnitude of M-errors and S-errors (*sensu Gelman & Carlin, 2014*, details below).

In all cases the simulated samples were defined by three parameters: the standard deviation of the sampled Gaussian distributions, the size of the samples, and the true effect size. The true effect is the value of the statistic of interest (*e.g.*, the t-value or the correlation coefficient) that would be observed in an infinitely large sample (*Gelman & Carlin, 2014*). In our simulations the true effect was defined by the parameters of the sampled Gaussian distributions (*e.g.*, the difference between the means of the two sampled Gaussian for t-test). We standardized the true effects on sample standard errors to make effects comparable across designs (*Lipsey & Wilson, 2001*). The expressions for these standardized true effect size (henceforth used interchangeably with "effect size" or simply "effect") for each analysis design are provided below in the descriptions of the simulations of each design.

We used Latin hypercube sampling to build 2,000 combinations of parameters and sample sizes from uncorrelated uniform distributions (*Chalom & Prado, 2016*). The sample sizes ranged from 10 to 100 and effect sizes and standard deviations ranged from 0.1 to 8. Thus, our combinations are hypercube samplings of parameter spaces that cover typical sample sizes of studies in ecology, and small to large effect sizes within a wide range of variation of data distributions. For each of these combinations we repeated the simulation 10,000 times. We also repeated the same procedures to run 10,000 simulations with 2,000 unique combinations of standard deviations and sample sizes for the case of zero effect size, in order to simulate a situation when the null hypothesis was true.

For every analysis simulation, we extracted the proportion of the simulations that yielded a correct conclusion. In the NHT approach, such measurements were the proportion of analyses that resulted in a p value lower than 0.05 if H0 was false, and higher than 0.05 otherwise. For the IT approach, we fit by maximum likelihood (*i.e.* by minimizing the sum of squares of residuals) the Gaussian models that express the null and alternative hypothesis for each design, as detailed below. We then considered a correct conclusion if the model that expressed the correct hypothesis was selected. To decide which model to select, however, we took into consideration the bias of AIC to select models
with uninformative parameters (*Teräsvirta & Mellin, 1986*). This problem arises when the true model is included in the selection procedure, along with models that provide the same fit but have additional uninformative parameters (*Aho, Derryberry & Peterson, 2014*). As this is the case in our simulations for ANOVA and linear regression (see below and in Appendices), we identified the model with fewer parameters that was among the models with $\Delta AIC < 2$ chosen using the IT approach (*Arnold, 2010*).

To estimate the S-error rate and M-error size (*Gelman & Carlin, 2014*) from each approach, we used the subset of simulations in which some effect was detected by NHT or IT. We estimated type-S error rate as the proportion of this subset in which the detected effect had the opposite sign of the true effect. The expected type-M error was estimated as the mean ratio between the estimated effects and the true effect value in the subset of simulations defined above.

In the appendices we have provided the functions in R (*R Development Core Team, 2016*) that we created to run the simulations and the R scripts of all simulations and analyses.

## t-test designs

We simulated an unpaired t-test design by drawing samples from two Gaussian distributions that differed in their means, but had the same standard deviation. One of the distributions had a mean of zero. The standardized effect size was the true t-value, which in this case is:

$$E_t = \frac{\mu}{\sigma\sqrt{\frac{2}{N}}} \tag{1}$$

where $\mu$ is the distribution mean which is allowed to be different from zero, and $\sigma$ and $N$ are the common standard deviations of both distributions and common samples sizes, respectively.

For the NHT approach, we calculated the t-value estimated from the samples and its corresponding *p*-value. For the IT approach, we calculated the Akaike Information Criterion corrected for small samples (AICc, *Burnham & Anderson, 2002*) of the two Gaussian linear models that express the null hypothesis and the alternative hypothesis. We recorded as correct conclusions of NHT the simulations in which $p < 0.05$ when $E_t \neq 0$ and the simulations in which $p > 0.05$ when $E_t = 0$. Accordingly, we recorded as correct conclusions of IT the simulations in which the model that expressed the correct statistical hypothesis was selected.

## ANOVA designs

We simulated three samples from Gaussian distributions to represent measures obtained from three experimental groups. All distributions had the same standard deviations, but the true mean of one of the distributions differed by a certain amount from the mean of the other two distributions, which was set to zero. We expressed the true standardized effect size in this case as an extension of the true t-value:

$$E_{ANOVA} = \frac{\mu}{\sigma\sqrt{\frac{3}{N}}} \qquad (2)$$

For the NHT approach, we used the F-test to test the null hypothesis. If the null hypothesis was rejected, a post-hoc Tukey's test was used. For the simulations when the true difference was not zero, the conclusion was considered correct only if the three $p$-values of the Tukey's test agreed with the simulation. No Tukey's test was used when the difference between means was set to zero. In those situations, a non-significant F-test was considered a correct conclusion and a significant one was considered a wrong conclusion.

For the IT approach we fit five linear Gaussian models to express all possible statistical hypotheses regarding the differences among the three experimental groups. Of these models, one had a single parameter representing the means of all the groups expressing the null hypothesis; three had two parameters for means, which allowed one group mean to be different from the other two; and one had a parameter for each group mean expressing the hypothesis that all group means differed. The values of AICc for each model were then calculated and we took as a correct conclusion the simulations in which the selected model agreed with the simulated situation.

## Correlation designs

For the correlation design, samples of paired variables were taken from a bivariate normal distribution with correlation parameter ranging from zero to positive values. The true standardized effect expression in this case was the correlation parameter expressed as a t-value (*Lipsey & Wilson, 2001*):

$$E_r = \rho\sqrt{\frac{N-2}{1-\rho^2}} \qquad (3)$$

where $\rho$ is the correlation of the bivariate Gaussian distribution from which the samples were drawn.

For the NHT approach, the $p$-value of the Pearson correlation coefficient of the two variables were calculated. For the IT approach, two models were fit. The first corresponded to the null hypothesis that the paired values come from a bivariate normal distribution with correlation parameter set to zero. The alternative hypothesis was represented by a model of a bivariate normal distribution with the correlation as a free parameter.

## Multiple linear regression designs

The multiple linear model designs had three variables: the response variable ($Y$), and two uncorrelated predictor variables ($X_1$ and $X_2$). The response variable was a linear function of $X_1$ plus an error sampled from a Gaussian distribution with zero mean and standard deviation $\sigma$:

$$Y = \beta_0 + \beta_1 X_1 + \varepsilon, \qquad \varepsilon \sim N(0, \sigma)$$

The standardized true effect size can thus be expressed as the standardized linear coefficient of $X_1$:

$$E_\beta = \frac{\beta_1}{\sigma} \sqrt{N} \qquad\qquad\qquad (4)$$

For the NHT approach we fitted a multiple linear regression including the additive effect of $X_1$ and $X_2$ and calculated the $p$-values of the Walt statistics to test the effect of each predictor. The probability of correct conclusions was estimated by the proportion of simulations that yielded a $p$-value for the $X_1$ corresponding to the correct hypothesis and a non-significant $p$-value for the $X_2$ variable. For the IT approach, four models were fit. The first was an intercept-only model where the expected value of the response $Y$ is constant. The other models included only the effect of $X_1$, only the effect of $X_2$, or the effects of both variables. The values of AICc for each model were then calculated and we took as a correct conclusion the simulations in which the selected model was in accordance with the simulated situation. To check the effect of collinearity, we repeated the simulations above forcing a correlation of 0.5 between $X_1$ and $X_2$. As the results did not show any important difference, we included this additional analysis in the appendices (See Section S2).

### Measuring evidence through *p*-values

We also explored the relationship between the $p$-value and the Akaike weight ($w$), which is proposed as a true measure of strength of evidence (*Burnham & Anderson, 2002*). Ultimately, we wanted to check if there is a pragmatic disadvantage in considering lower $p$-values as "less evidence of the null hypothesis". A monotonic positive relationship between the $p$-values and the evidence weights for the model that express the null hypothesis ($w_{H0}$) would imply no pragmatic disadvantage. To check the relationship between the $p$-value and $w_{H0}$, we recorded both values for each simulation, for all four designs. It is important to emphasize that interpreting $p$-values as strength of evidence is conceptually wrong (*Anderson & Burnham, 2002*; *Dennis et al., 2019*). However, our main interest here is to understand if interpreting $p$-values as such, although conceptually wrong, has any pragmatic disadvantage.

## RESULTS

### Significance, power, S-errors and M-errors

When the null hypothesis was correct, the NHT approach achieved the nominal probability of type-I error ($\alpha = 0.05$) for the t-test, ANOVA, and correlation designs and a value close to $\alpha = 0.1$ for linear regression. The IT approach performed slightly better in the t-tests, correlation and linear regression (Table 1).

When the null hypothesis was wrong, the average proportion of correct conclusions in the simulations was used to estimate the test power $\beta$. For all cases where the NHT

**Table 1 Proportions of type-I error in the simulations, for the Null Hypothesis Tests (NHT) and information-based model selection (IT).**

|  | NHT | IT |
|---|---|---|
| t-test | 0.050 | 0.044 |
| Correlation | 0.050 | 0.012 |
| ANOVA | 0.050 | 0.112 |
| Regression | 0.097 | 0.091 |

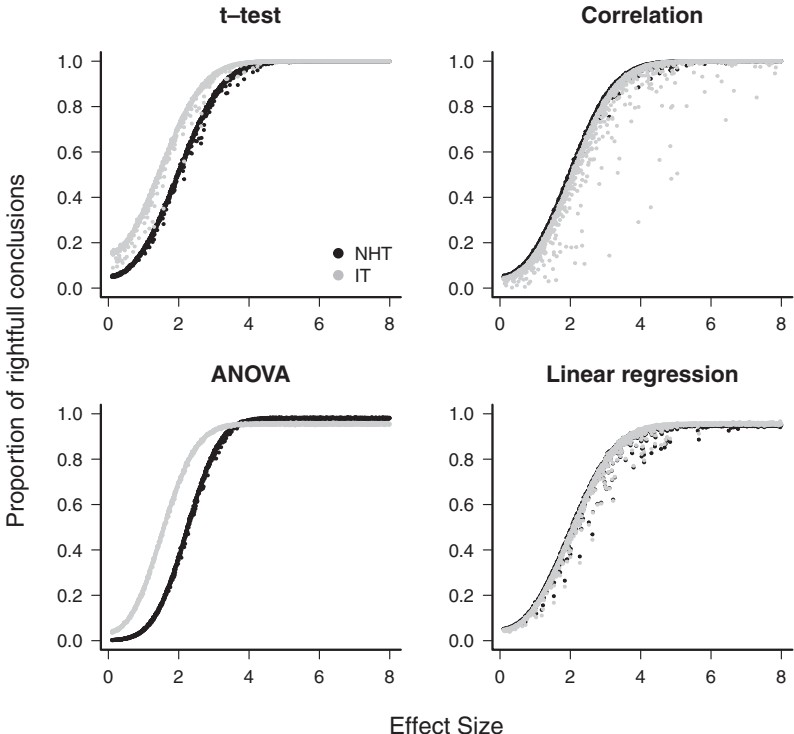

**Figure 1 The power of null hypothesis tests (NHT, black) and the information-based model selection (IT, grey) for each testing design, as a function of effect size.** Each point is the proportion of the 10,000 simulations of a test instance from which the effect was detected. Each test instance used a different combination of effect size, standard deviation of the values and sample size.

approach was used, the power was less than $\beta = 0.2$ for effect sizes below one, and achieved $\beta = 0.8$ for effect sizes of about 2.8 (Fig. 1), as expected for the Gaussian distribution (*Gelman & Carlin, 2014*). The IT approach produced larger estimated power for small effect sizes in the t-test and ANOVA designs, but in all cases the power of both approaches converged to $\beta \approx 1$ as the effect size approaches 4.0 (Fig. 1). The IT approach only achieved this convergence with the additional parsimony criteria to discard models with uninformative parameters (see Supplemental Information S2).

   The exaggeration rate or type-M error was at least 2.0 when effect sizes were about 1.5 standard error units for all test designs and approaches (Fig. 2), and M-errors increased

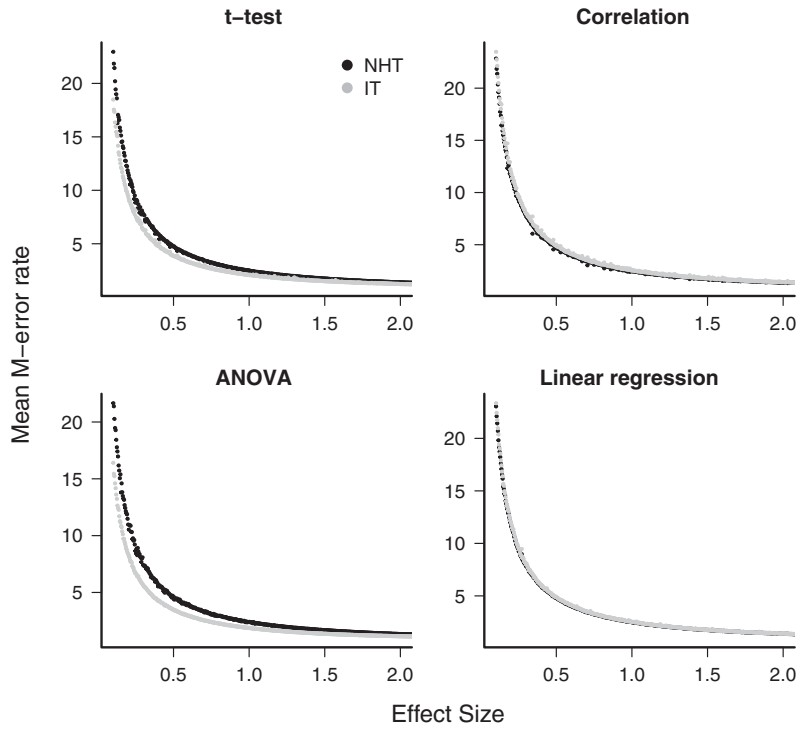

**Figure 2 The mean type-M error (exaggeration rate) of null hypothesis tests (NHT, black) and information-based model selection (IT, grey) for each testing design, as a function of effect size.** Each point represents the simulations of a test instance from which an effect was detected. Each test instance used a different combination of effect size, standard deviation of the values and sample size and was simulated 10,000 times. The M-error is the absolute ratio between the estimated effect size and the true effect size (*Gelman & Carlin, 2014*), which was estimated from the mean of this ratio for each test instance.                                

steeply as the effect size decreased. Thus when an effect below 1.5 was detected, the true value was exaggerated at least twofold. The IT approach had a slightly lower type-M error than NHT for the t-test and ANOVA designs for effect sizes below 2. Type-S error decreased more abruptly than the M-error with the increase of effect size (Fig. 3). In our simulations the probability that the detected effect is of the wrong sign was of some concern (larger than 0.1) for effect sizes well below unit, but the IT approach had slightly larger S-errors at this range than the NHT for the t-test and ANOVA designs (Fig. 3). In all test designs and both approaches, S and M errors vanished for effect sizes greater than 1.5 and 2.0, respectively. Collinearity in the linear regression design did not change none of the patterns described above (Supplemental Information S2).

## *p*-value as a measure of strength of evidence

The relationship between *p* and evidence weight *w* was positive and monotonic as expected. Simulations with larger effect sizes resulted in a small *p*-value and small evidence weight for the null hypothesis. Within the range of $0 < p < 0.1$, there was little variation around the trend, despite the wide range of parameters sampled by the hypercube and used in the simulations (Fig. 4).

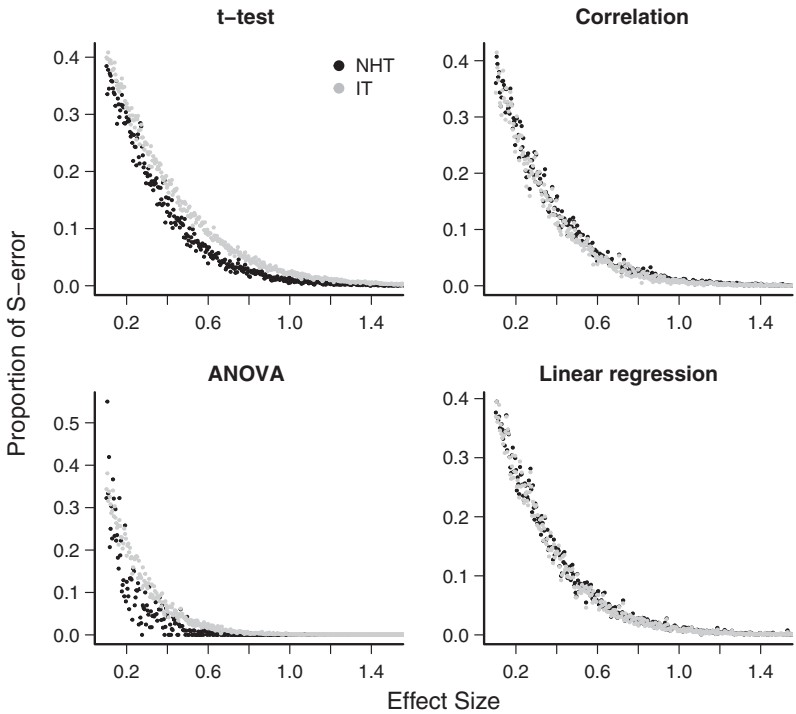

**Figure 3 Mean type-S error of Null Hypothesis Tests (NHT, black) and information-based model selection (IT, grey) for each testing design, in function of effect size.** Each point represents the simulations of a test instance from which an effect was detected. Each test instance used a different combination of effect size, standard deviation of the values and sample size and was simulated 10,000 times. The S-error is the probability of detecting an effect of an opposite sign of the true effect (*Gelman & Carlin, 2014*). For each test instance we estimated S-errors from the proportion of simulations that detected an effect of the opposite sign.

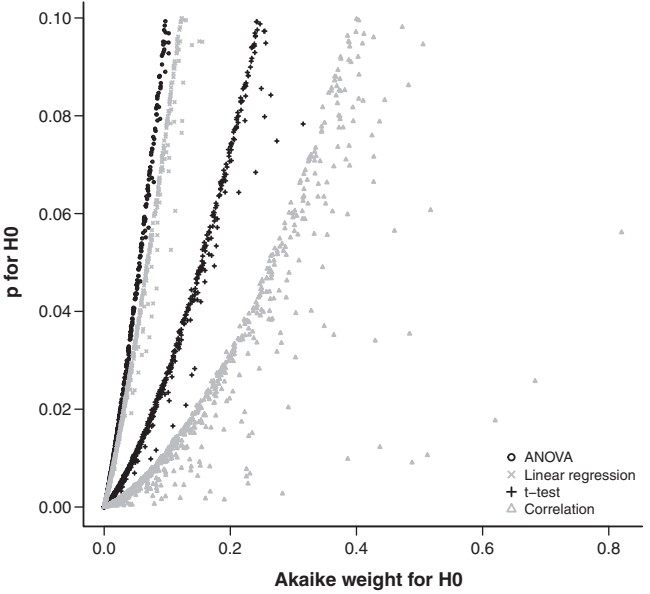

**Figure 4 Relationship between the Akaike weight of the null model and the *p*-value of the null hypothesis found in simulations of each analysis design.**

## DISCUSSION

### Comparing the performance of the two approaches

For all designs we have simulated, the response of the power, the S-error and M-error to the standard effect size were very similar in the NHT and IT approaches. The increase in power with effect size for NHT is well known and expected from the consistency of estimators of the Gaussian distribution (*e.g. Edwards, 1972*) behind these tests. It had been also demonstrated using likelihood-ratio tests approximations (*Aho, Derryberry & Peterson, 2017*, see also Supplemental Section S1). The lack of consistency of *AIC* when alternative models with uninformative parameters are considered has been highlighted recently (*Aho, Derryberry & Peterson, 2014*, *2017*; *Dennis et al., 2019*; *Leroux, 2019*; *Tredennick et al., 2021*). This characteristic is due to AIC being created mostly with prediction in mind (*Dennis et al., 2019*). This, however, was easily circumvented with the additional parsimony criteria proposed by *Arnold (2010)*. The relationship between S and M errors to test power (and thus to effect size) has, to date, received far less attention. We have shown that this relationship for all four of the Gaussian designs simulated agrees with those predicted by approximating the distribution of effects to a t-distribution (*Gelman & Carlin, 2014*). As the standard effect size (and power) increases, both S and M errors decrease steeply, but S errors are a concern for effect sizes below one standard error, which in our simulations correspond to a test power between 0.05 (ANOVA, NHT) to 0.34 (t-test, IT). Accordingly, the exaggeration rate (M error) of twice or more occurred when the effect size is below 1.5 standard error units, which corresponds to a power value between 0.16 (ANOVA, NHT) and 0.53 (t-test, IT). These results also showed that the greater power and lower M-error mean rate of IT at small effect sizes for the t-test and ANOVA come at the cost of an increased ratio of S-error.

In summary, the performance of IT and NHT were very similar and converged quickly as standard effect sizes increased, which is caused by an increase in raw effect sizes, sample sizes or a decrease in standard errors. All these factors increase power, and is largely recognized that different inference criteria lead to the same conclusions as power increases (*Gelman & Carlin, 2014*; *Ioannidis, 2005*; *Button et al., 2013*). Focusing on how to obtain the best estimates of the effects can thus be a more effective contribution to scientific advancement than to dispute the value of weak inferences obtained with different statistical approaches (*e.g. Gelman & Loken, 2014*).. Effect estimates can be improved by increasing sample sizes or controlling error sources. Our results suggest that test designs should target a standard effect size of 2.8, which correspond to a test power above 0.8.

Moreover, statistical inference relies on the full sequence of events and decisions that led to the conclusions presented, which is not just the statistics used nor their power in a given sampling or experimental design (*Gelman & Loken, 2014*; *Greenland et al., 2016*). The choice of different inference approaches is only a part of this problem but it has dominated the debate. It might be the time to broaden our concerns to address the whole path that leads from a research hypothesis to a statistical hypothesis to be evaluated (*Gelman, 2013*). Finally, we did not address the issue of the biological relevance of an effect that was correctly detected, because we assume this task is beyond the purpose of statistics

and should be left to researchers. In many situations, procedures such as model averaging and effect size statistics can also be used to enhance the predictive power of models and to support the process of drawing conclusions, and also help reduce type S-error and type M-error rates. However, assessing how such *post hoc* procedures could improve a statistical method is beyond the scope of this article. Here, we deal with the initial values guiding drawing conclusions (*i.e.*: p-values and AICcs), as any statistical procedure done at a later stage would be conditional on those values. Those interested in how model averaging can enhance predictive power and how this relates to more traditional techniques are directed to *Freckleton (2011)*.

### p-value as a measure of strength of evidence

One of the strongest arguments recently given by the major proponents of the IT approach against the traditional NHT is that the *p*-value is not a measure of strength of evidence in favor of the null hypothesis. Nevertheless, there is a widespread interpretation of *p*-values as "more significant" (*i.e.* less supportive of the null hypothesis) the lower they get. Furthermore, although an ecologist would hardly discard the null hypothesis based on a *p*-value higher than 0.1, values between 0.1 and 0.05 are usually interpreted as moderate evidence against the null hypothesis (*Murtaugh, 2014*). The relationship between the *p*-value and other statistics taken from the IT approach has been demonstrated before for the case of nested models in which the sample size is large enough to apply the log-likelihood ratio test (LRT) (*Murtaugh, 2014*; *Greenland et al., 2016*). We extended this conclusion for the simple designs we evaluated without the assumptions of LRT. The relationship between *p*-value and Akaike weights is monotonic positive and was poorly affected by variations of the simulations, specially at the borderline of significance.

Other authors have already pointed out that for many simple cases there is a monotonic relationship between *p*-values and likelihood ratios and thus to evidence weights (*Edwards, 1972*; *Royall, 2000*), by translating standard significance tests into alternative models with different parameter values (*e.g.* Fig. S2). Therefore we argue that the interpretation of *p*-values as measures of evidence, although conceptually wrong (*Edwards, 1972*; *Cohen, 1994*; *Royall, 2000*), can be empirically useful at least for the standard significance test designs.

### Concluding Remarks

We compared null hypothesis testing and information-theoretic approaches in situations commonly found by ecologists, considering sample sizes and correlation degrees often reported in ecological studies and only focusing on the important practical issues of both methods. The few differences between IT and NHT showed a trade-off between M and S errors and vanished as the effect size increases. We also showed that, at the borderline of significance in standard procedures with Gaussian errors, *p*-values can be used as a very good approximation of a measure of evidence to the null hypothesis when compared to the alternative.

The recent statement that NHT is an outdated method for analyzing data is not supported by our findings. The basic NHT designs we analyzed have been the basis of data

analyses for generations of ecologists, and still prove to be valuable in the context they were created (*Gelman, 2013*; *Stanton-Geddes, De Freitas & De Sales Dambros, 2014*). Many criticisms to NHT are valid, but the IT approach has also been correctly criticized and some of those criticisms are even the same as the ones used to justify NHT as an outdated technique (*Arnold, 2010*; *Freckleton, 2011*; *Hegyi & Garamszegi, 2011*; *Richards, Whittingham & Stephens, 2011*). Besides, for those uncomfortable with the NHT technique, the IT technique is not the only alternative. Several alternative methods, all with their own pros and cons, have been proposed (*Hobbs & Hilborn, 2006*; *Garamszegi et al., 2009*), including other indexes derived from the information theory, like the well known Bayesinan Information Criterion (BIC), which has a different assyntotic behavior than AIC, being statistically consistent, rather than efficient (*Dennis et al., 2019*).

As in any simulation study, the generality of the differences found in the performance of NHT and IT cannot be afforded beyond the parameter space that we have explored. Nevertheless, the simulations show that insisting on the absolute supremacy of a given approach is pointless, at least from a pragmatic perspective that seeks agreements in the findings despite the statistics used. We have shown a simple instance of agreement in which two statistical approaches in many situations lead to the same conclusions. In this case we elucidated the causes of the few divergences found, as well as the simple mathematical relationships that explain the concordances. Whether agreements are also possible by other means and in more complex designs remains to be evaluated. Nevertheless, by focusing on the consequences of a given result, a pragmatic view of statistics has a greater potential to find a common ground from different pieces of evidence and to promote a more insightful dialogue between researchers.

## ACKNOWLEDGEMENTS

Our thanks to Eduardo S. A. Santos and Tadeu Siqueira for their keen suggestions to previous versions of this paper.

### Funding

Paulo Inácio Prado has a research grant 2013/19250-7 of São Paulo Research Foundation (FAPESP) and a scientific productivity grant from CNPq. The funders had no role in study design, data collection and analysis, decision to publish, or preparation of the manuscript.

### Grant Disclosures

The following grant information was disclosed by the authors:
Paulo Inácio Prado: 2013/19250-7.
CNPq.

### Competing Interests

The authors declare that they have no competing interests.

## Author Contributions

- Leonardo Braga Castilho conceived and designed the experiments, performed the experiments, analyzed the data, authored or reviewed drafts of the paper, and approved the final draft.
- Paulo Inácio Prado conceived and designed the experiments, performed the experiments, analyzed the data, prepared figures and/or tables, authored or reviewed drafts of the paper, and approved the final draft.

## Data Availability

The R codes are available in GitHub: https://github.com/piklprado/NHTxIT.

## Supplemental Information

Supplemental information for this article can be found online at http://dx.doi.org/10.7717/peerj.12090#supplemental-information.

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
