# Peer review of "Towards a pragmatic use of statistics in ecology"

_PeerJ, doi:10.7717/peerj.12090_

## Round 0.1 · original submission · Major Revisions

· Academic Editor

Major Revisions

Three consistent reviews have been received. Reviewers find that the paper is interesting and potentially publishable. However, there are some concerns in the technical methodology. Please prepare a detailed point-to-point response to the reviewers' questions.

Reviewer 1 ·

Basic reporting

See my general comments

Experimental design

See my general comments

Validity of the findings

See my general comments

Additional comments

This is an interesting study and well-designed simulation work. I have just three comments

1) Given that this paper is for ecologists, the authors may have missed a couple of papers that dealt with Type M and S errors:

Lemoine, Nathan P., et al. "Underappreciated problems of low replication in ecological field studies." Ecology 97.10 (2016): 2554-2561.

Cleasby, Ian R., et al. "What is our power to detect device effects in animal tracking studies?." Methods in Ecology and Evolution (2021).

These publications should probably be acknowledged as they do deal with Type M and S errors.

2) I disagree with the use of "Effect size" in this paper. What the authors are calling 'effect size' is t values or z values (this use is unusual and even misleading). They are usually not considered so - see this Wiki page and references therein

https://en.wikipedia.org/wiki/Effect_size

I think they should use effect size such as Cohen's d (also known as standardized mean difference), which can be obtained for correlations via conversation.

3) Relating to my Point 2, I think the simulation should show the relationships among effect size, sample size and NHT (or IT). I think this will be much more meaningful than what is presented currently. Under their "effect size", the effect and sample sizes are confounded and I do not think it is very useful and not informative.

Reviewer 2 ·

Basic reporting

No comment.

Experimental design

No comment.

Validity of the findings

The findings are interesting and well-supported.

Additional comments

The literature review should be significantly improved with summaries of recent studies (preferably after 2019)

Reviewer 3 ·

Basic reporting

The article is well written and clear. My one comment is that I suggest moving the last paragraph of the introduction, where you discuss some caveats of the work, to the discussion.

Experimental design

I found the topic and method of research to be very interesting. The debate between IT and NHST approaches has been an ongoing topic of discussion, and the author's simulations make an important contribution. Their R code all appears to be solid to me, and their general methods of comparing the two methods makes sense.

A potential aspect of the IT approach that I feel was somewhat missed out is that multimodel inference allows for quantification of measures of "variable importance" (often summarized as the sum of model weights for all models containing each variable) and model averaging of coefficients, which may have predictive benefits in some situations. I don't think these concepts need to necessarily be included in the simulations, however model averaging specifically might (I emphasize *might*) reduce the type-M error for the IT approach and might therefore bear mentioning at least.

I'd also be interested in how other information criteria perform. As the authors say, AIC doesn't assume that the true model is in the candidate set, which makes it vulnerable to overfitting in situations where the true model is one of the candidates. How might other criteria, like BIC, or truly Bayesian methods like WAIC, perform? Again, I don't think its necessary that these be included in the simulations themselves, as I think that the current design is quite elegant and well defined, but it would be good to mention them as possibilities.

Validity of the findings

The findings of this study are quite interesting, and I think will be useful for many researchers designing their statistical analysis. I think the fundamental point that we should focus more on designing experiments or observational protocols that allow us to easily detect and quantify "true" effect, rather than the relatively petty disagreement around IT vs NHST, is a very important one that I'm happy to see be made.

My one area of concern is in their discussion of p-values as a measure of evidence weight. While their simulations do show that, in these specific cases, there is a monotonic positive relationship between p-values and the weight of the null model, I'd emphasize, as the authors themselves state, that interpreting the p-values in this way is conceptually incorrect. While I understand their point, that previous studies that interpret p-values in this way are likely to have come to similar conclusions as they would have if they measured model weights, I worry that this could be read as encouraging future studies to use p-values in this way. If possible, I'd like the authors to clarify what their position is here, and potentially amend the text to more clearly indicate that while it can serve as a rough approximation, interpreting p-values as the likelihood of the null hypothesis is fundamentally incorrect.

Additional comments

This is an important and interesting simulation study, well conceived and reported. I look forward to hearing the authors responses to the few queries I've raised here.

---

## Round 0.2 · accepted · Accept

· Academic Editor

Accept

Reviewers' comments have been addressed. I recommend acceptance of the paper.

Reviewer 1 ·

Basic reporting

I still think the use of Cohen's d or similar would be more in line with what I am used to and (probably more general and useful). However, I do see the authors' viewpoint as well. It is true many ecologists and evolutionary biologists are more familiar with t and z values rather than d. Given this, I have no further comments.

I believe the paper is clearly written with good simulation study (design) and it is great that all materials are open and accessible.

Experimental design

See above

Validity of the findings

See above

Additional comments

no comment